# Norm-guided latent space exploration for text-to-image generation

**Dvir Samuel**[1,2]**, Rami Ben-Ari**[2]**, Nir Darshan**[2]**, Haggai Maron**[3,4]**, Gal Chechik**[1,4]
[1]Bar-Ilan University, Ramat-Gan, Israel
[2]OriginAI, Tel-Aviv, Israel
[3]Technion, Haifa, Israel
[4]NVIDIA Research, Tel-Aviv, Israel

## Abstract

Text-to-image diffusion models show great potential in synthesizing a large variety of concepts in new compositions and scenarios. However, the latent space of initial seeds is still not well understood and its structure was shown to impact the generation of various concepts. Specifically, simple operations like interpolation and finding the centroid of a set of seeds perform poorly when using standard Euclidean or spherical metrics in the latent space. This paper makes the observation that, in current training procedures, diffusion models observed inputs with a narrow range of norm values. This has strong implications for methods that rely on seed manipulation for image generation, with applications to few-shot and long-tail learning tasks. To address this issue, we propose a novel method for interpolating between two seeds and demonstrate that it defines a new non-Euclidean metric that takes into account a norm-based prior on seeds. We describe a simple yet efficient algorithm for approximating this interpolation procedure and use it to further define centroids in the latent seed space. We show that our new interpolation and centroid techniques significantly enhance the generation of rare concept images. This further leads to state-of-the-art performance on few-shot and long-tail benchmarks, improving prior approaches in terms of generation speed, image quality, and semantic content.

## 1 Introduction

Text-to-image diffusion models demonstrate an exceptional ability to generate new and unique images. They map random samples (seeds) from a high-dimensional space, conditioned on a user-provided text prompt, to a corresponding image. Unfortunately, the *seed space*, and the way diffusion models map it into the space of natural images are still poorly understood. This may have a direct effect on generation quality. For example, diffusion models have difficulty generating images of rare concepts, and specialized methods have been proposed to resolve this issue, for example, by performing optimization in the seed space [43]. Our limited understanding of the seed space is further demonstrated by the fact that standard operations on seeds, such as interpolating between two seeds or finding the centroid of a given set of seeds, often result in low-quality images with poor semantic content (Figure 1 left, first two rows). As a result, methods based on the exploration and manipulation of seed spaces face a considerable challenge.

The aim of this paper is to propose simple and efficient tools for exploring the seed space and to demonstrate how these tools can be used to generate rare concepts. Our main observation is that a specific property, the norm of the seed vector, plays a key role in how a seed is processed by the

---

*Correspondence to: Dvir Samuel <dvirsamuel@gmail.com>

37th Conference on Neural Information Processing Systems (NeurIPS 2023).

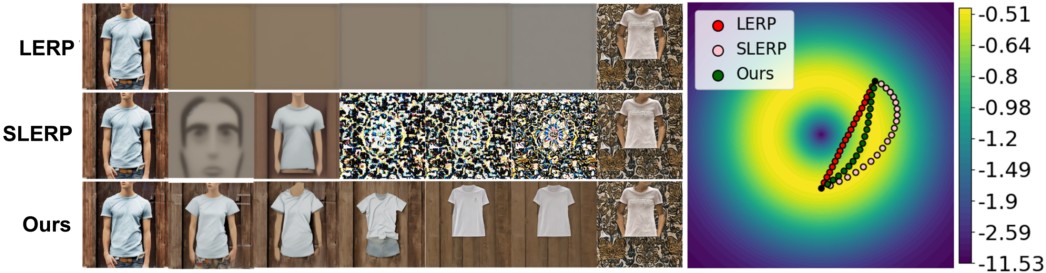

Figure 1: (Left) Visual comparison of different interpolation methods between two seeds from high-dim space of StableDiffusion [38]. Images generated using Stable Diffusion [38]. (Right) Paths found using Linear, Spherical, and likelihood-based interpolation methods in 2D space, where the norm of samples has a $\chi$ distribution (log) PDF. Note that NAO interpolation path goes through the domain with the desired norm (yellow-green). Both linear interpolation and SLERP [48] produce seeds that result in images that are flat or just noise, whereas our approach is capable of generating images that are meaningful (quantified in Table 1).

diffusion model. In more concrete terms, since seeds are sampled from a multidimensional Gaussian distribution, the norm of the seeds has a $\chi$ distribution. For high dimensional Gaussian distributions, such as the ones used by diffusion models, the $\chi$ distribution is strongly concentrated around a specific positive number. Consequently, diffusion models tend to favor inputs with this norm, resulting in lower-quality images when the norm of the seed differs from it.

To address this issue, we propose to use a prior distribution over the norms in the seed space based on the $\chi$ distribution to guide exploration. While prior-based exploration techniques for seed spaces have been proposed before [2, 4], the advantage of our prior is that it does not rely upon an expansive estimation of the empirical data distribution nor on complex computations and hence can be applied to very high dimensional latent spaces. Yet, as we show below, our prior still significantly improves exploration techniques in seed space.

As a first step, we propose a novel method for interpolating between two seeds. In contrast to Linear Interpolation (LERP) or Spherical Linear Interpolation (SLERP) [48], we formulate this problem as finding a likelihood-maximizing path in seed space according to the aforementioned prior. In addition to providing us with an interpolating path, we also demonstrate that the optimal value of this optimization problem defines a new non-Euclidean metric structure over the seed space. Figure 1 compares our interpolation paths to two other frequently used interpolation methods in 2D and in image space. The improvement of the image quality along the path is evident. Specifically, the 2D example (right panel) illustrates that LERP and SLERP paths cross low-probability areas whereas our path maintains a high probability throughout. The same phenomenon is shown for images (left panel) where it is apparent that intermediate points in the paths generated by the baseline methods have a significantly lower quality.

As a next step, we build on our newly defined metric to define a generalized centroid for a set of seeds. In contrast to the standard definition of the centroid in Euclidean spaces, we define the centroid as the point that minimizes the distances to the seeds according to the new distance function (also known as the Fréchet mean for that given metric). We show how to discretize the two optimization problems above and solve them using a simple and efficient optimization scheme. We call our approach **NAO** for *Norm-Aware Optimization*.

We evaluate NAO extensively. First, we directly assess the quality of images generated by our methods, showing higher quality and better semantic content. Second, we use our seed space interpolation and centroid finding methods in two tasks: (1) Generating images of rare concepts, and (2) Augmenting semantic data for few-shot classification and long-tail learning. For these tasks, our experiments indicate that seed initialization with our prior-guided approach improves SoTA performance and at the same time has a significantly shorter running time (up to X10 faster) compared to other approaches.

## 2   Related Work

**Text-guided diffusion models.** Text-guided diffusion models involve mapping random seed (noise) $z_T$ and textual condition $P$ to an output image $z_0$ through a denoising process [7,36,41]. The inversion of this process can be achieved using a deterministic scheduler (e.g. DDIM [50]), allowing for the recovery of the latent code $z_T$ from a given image $z_0$. See a detailed overview in the supplemental.

**Rare concept generation with text-to-image models.** Diffusion models excel in text-to-image generation [7, 36, 41], but struggle with rare fine-grained objects (*e.g.*payphone or tiger-cat in StableDiffusion [38]) and compositions (*e.g.*shaking hands) [28,43]. Techniques like pre-trained image classifiers and text-driven gradients have been proposed to improve alignment with text prompts, but they require pre-trained classifiers or extensive prompt engineering [17, 22, 29, 33, 34, 41, 57]. Other approaches using segmentation maps, scene graphs, or strengthening cross-attention units also face challenges with generating rare objects [5, 10, 18, 19, 63]. SeedSelect [43] is a recent approach that optimizes seeds in the noise space to generate rare concepts. However, it suffers from computational limitations and long generation times. This paper aims to address these limitations by developing efficient methods that significantly reduce generation time while improving the quality of generated images.

**Latent space interpolation.** Interpolation is a well-studied topic in computer graphics [21,49,51]. Linear Interpolation (LERP) is commonly used for smooth transitions by interpolating between two points in a straight line. Spherical Linear Interpolation (SLERP) [48], on the other hand, offers computing interpolation along the arc of a unit sphere, resulting in smoother transitions along curved paths. Image interpolations in generative models are obtained by three main approaches: (1) Linear or spherical interpolation between two latent vectors [1,37,47,65], (2) Image-to-Image translation approaches [31] and (3) Learning an interpolation funciton or metrics based on the data [4, 11, 25, 47]. [3] observed that linearly traveling a normally distributed latent space leads to sub-optimal results and proposed an interpolation based on a Riemannian metric. [2] further proposed a methodology to approximate the induced Riemannian metric in the latent space with a locally conformally flat surrogate metric that is based on a learnable prior. Note, that as opposed to our approach, these priors do not have a closed-form solution, they work on a relatively low dimensional latent space of a VAE (constrained and compact latent space), and they learn the metric from the data itself. In this paper, we do not assume any of the above. We introduce a novel interpolation approach that effectively use the inherent structure of the latent space to achieve correct interpolation without any additional data, on the high-dimensional seed space of a diffusion model.

**Data Augmentation via Latent Space Exploration.** Previous methods, such as [16, 30] and [6], have proposed techniques for semantic data augmentation using the latent space of generative models. These methods involve imposing uniform latent representations and applying linear interpolation or learning mappings to sample specific areas in the latent space. However, these approaches require training generative models from scratch. In contrast, this paper demonstrates a more efficient approach by utilizing the latent space of a pre-trained diffusion model for creating data augmentations without the need for additional model fine-tuning.

## 3   A norm-based prior over seeds

We start with reviewing statistical properties of samples in a seed latent space $z_T \in \mathbb{R}^d$, with $d$ denoting the dimension of the space.

In diffusion models, it is common to sample $z$ from a high-dimensional standard Gaussian distribution $z_T \sim \mathcal{N}(0, I_d)$. For such multivariate Gaussians, the $L_2$ norm of samples has a $\chi$ (Chi) distribution: $||z_T|| = \sqrt{\sum_{i=0}^{d} z^{i^2}} \sim \chi^d = ||z_T||^{d-1} e^{-||z_T||^2/2}/(2^{d/2-1}\Gamma(\frac{d}{2}))$ where $\Gamma(\cdot)$ is the Gamma function and $|| \cdot ||$ is the standard Euclidean norm.

Importantly, as the dimension grows, the distribution of the norm tends to be highly concentrated around the mean, since at a high dimension, the variance approaches a constant 0.5. This strong concentration is illustrated in the inset figure on the right, for $d = 16384 = 128^2$, the seed dimension used by Stable Diffusion [38]. At this dimension, the mean is

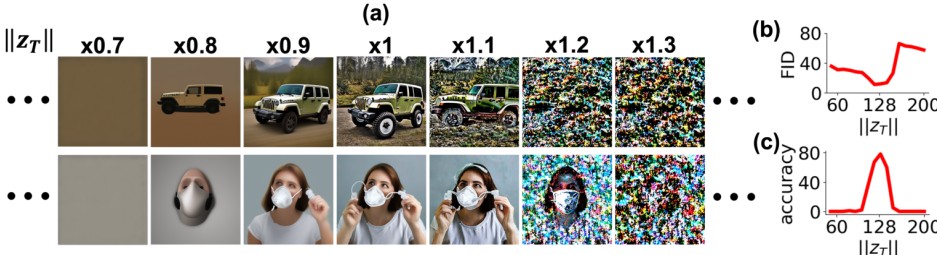

Figure 2: **(a)** Progressively changing the norm of a fixed seed, which initially has a norm of $\sqrt{d} = 128$. Images are generated by Stable Diffusion [38]. The visual quality of generated images degrades as the norm diverges away from $\sqrt{d}$ at the left and right end. **(b)** Mean per-class FID of the generated images in relation to the seed norms (lower is better). **(c)** Mean per-class accuracy of the generated images, as determined by a state-of-the-art pre-trained classifier, as a function of the seed norm. Plots (b) and (c) show the degradation of the image quality as the seed norm moves away from $\sqrt{d}$

also very close to the mode of the distribution, and both approach $\sqrt{d} = 128$. This property means that samples drawn from a multi-variate high-dimensional Gaussian distribution are concentrated around a specific value $r = \text{mode}(\chi^d) \approx \sqrt{d}$. Our key observation is that diffusion models are trained with inputs sampled from the above normal distribution, and therefore the models are only exposed to inputs with norm values close to $r$ during training. We hypothesize that this causes the model to be highly biased toward inputs with similar norm values.

We conducted several experiments to validate this bias. First, we inspected the visual quality of images generated with different norm values, all sharing the same direction of the seed vector. Figure 2(a) visually illustrates the sensitivity of the Stable Diffusion model to the input norm, showing that quality degrades as the norm drifts away from the mode. Second, we conducted a systematic quantitative experiment and measured the impact of seed norm on image quality in terms of FID and classification accuracy scores using an ImageNet1k pre-trained classifier. Figures 2(b)-2(c), show again that image quality depends on the seed having a norm close to the mode. Full details of these experiments are given in Section 5 and supplemental material.

We conclude that the norm of the seed constitutes a key factor in the generation of high-quality images. Below we describe how this fact can be used for seed optimization and interpolation.

## 4 Norm-guided seed exploration

Based on the above results, we define a prior over the seed space as $\mathcal{P}(z_T) := \chi^d(\|z_T\|)$. This probability density function represents the likelihood of a seed with norm $\|z_T\|$ to be drawn from the Gaussian distribution. We now describe simple and efficient methods for seed interpolation and centroid finding using that prior.

### 4.1 Prior induced interpolation between two seeds

We first tackle the task of finding an interpolation path between the seeds of two images. The derivation of this interpolation path illustrates the advantages of using the prior in a simple setup, and will also be used later for finding centroids for sets of seeds. As seen in Figure 1 (see also Figure 3), a linear interpolation path between seeds consists of seeds that yield low-quality images. Instead, we define a better path $\gamma : [0, 1] \to \mathbb{R}^d$ as the solution to the following optimization problem: Given two images, $I_1$ and $I_2$, and their corresponding inversion seeds, $z_T^1$ and $z_T^2$, derived by inversion techniques (e.g. DDIM Inversion [50]), we aim to maximize the log-likelihood of that path under our prior, defined as the line-integral of the log-likelihood of all points on the path.

Equivalently, we minimize the negative log-likelihood of the path, which is strictly positive, yielding

$$\inf_{\gamma} \; -\int_{\gamma} \log \mathcal{P}(\gamma)ds \quad \text{s.t.} \quad \gamma(0) = z_T^1, \gamma(1) = z_T^2. \tag{1}$$

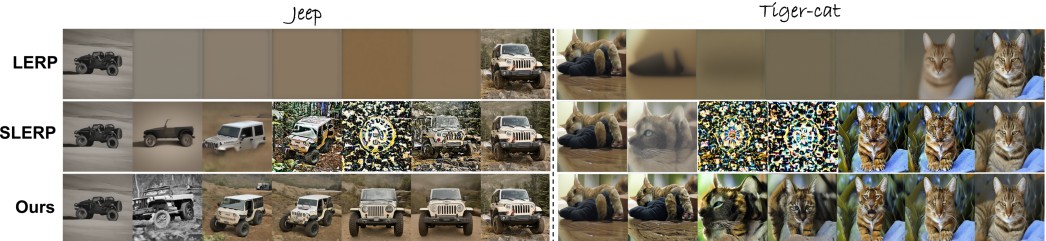

Figure 3: Qualitative comparison among different interpolation methods between two image seeds. "Jeep" is a common concept while "Tiger cat" is a rare concept. Images generated using SD [38].

Here, the infimum is taken with respect to all differentiable curves $\gamma$ and $\int_\gamma W(\gamma)ds$ denotes the line integral of a function $W : \mathbb{R}^d \to \mathbb{R}$ over the curve $\gamma$ [1]. We denote the optimal value obtained for optimization problem (1) as $f(z_T^1, z_T^2)$. It turns out that $f$ defines a (non-euclidean) distance on the seed space. This is stated formally in the following proposition.

**Proposition 1.** *For any distribution yielding strictly positive negative log-likelihood, and specifically when $\mathcal{P}$ is the $\chi^d$ distribution, then $f(\cdot, \cdot)$ is a distance function on $\mathbb{R}^d$.*

See the supplementary material for proof. It is important to note that the optimization problem does not only provide us with a path that maximizes the log-likelihood of our prior, but it also defines a new metric structure on the seed space that will prove useful in Section 4.2.

To approximate the solution to problem (1) in practice, we discretize the path into a sequence of piece-wise linear segments, that connect a series of points $z_T^1 = x_0, \ldots, x_n = z_T^2$ and replace the integral with its corresponding Riemann sum over that piece-wise linear path:

$$\underset{x_0,\ldots,x_n}{\text{minimize}} \quad -\sum_{i=1}^n \log \mathcal{P}\left(\frac{x_i + x_{i-1}}{2}\right)\|x_i - x_{i-1}\| \tag{2}$$
$$\text{s.t.} \quad x_0 = z_T^1, x_n = z_T^2, \quad \mathcal{C}(x) = |x_i - x_{i-1}\| - \delta \le 0, i \in \{1, \ldots, n\}$$

To facilitate a good approximation of the continuous integration, we also constrain consecutive path points to be close (see implementation at the end of Sec. 4.2).

Figures 1-3 illustrate paths resulting from the optimization of the discretized optimization problem. Our optimized path consistently produces higher-quality images compared to other methods. A quantitative evaluation is given in Section 5.

## 4.2   Prior induced centroid

Having defined a new metric structure in seed space, we are now ready to tackle the problem of finding a centroid of multiple seeds. To this end, we assume to have a set of images $\{I_1, I_2, \ldots, I_k\}$ with their inversions $\{z_T^1, z_T^2, \ldots, z_T^k\}$ and we wish to find the centroid of these images. For example, one possible use of such a centroid would be to find a good initialization for a seed associated with a rare concept based on a few images of that concept. This enables us to merge information between seeds and generate a more reliable fusion of the images that would not be achievable through basic interpolation between two seeds.

Recall that the centroid of a set of points is defined as the point that minimizes the sum of distances between all the points. Perhaps the simplest way to define a centroid in our case is by using the Euclidean distance, where the

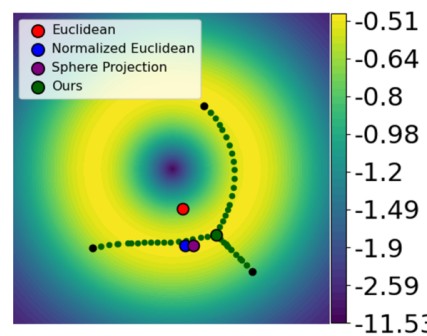

Figure 4: Comparing different centroid finding methods in 2D space on the contour of the $\chi$ distribution (log) PDF.

---

[1]When $\gamma$ is differentiable, the integral can be calculated using the following formula: $\int_0^1 W(\gamma(t))\|\gamma'(t)\|dt$

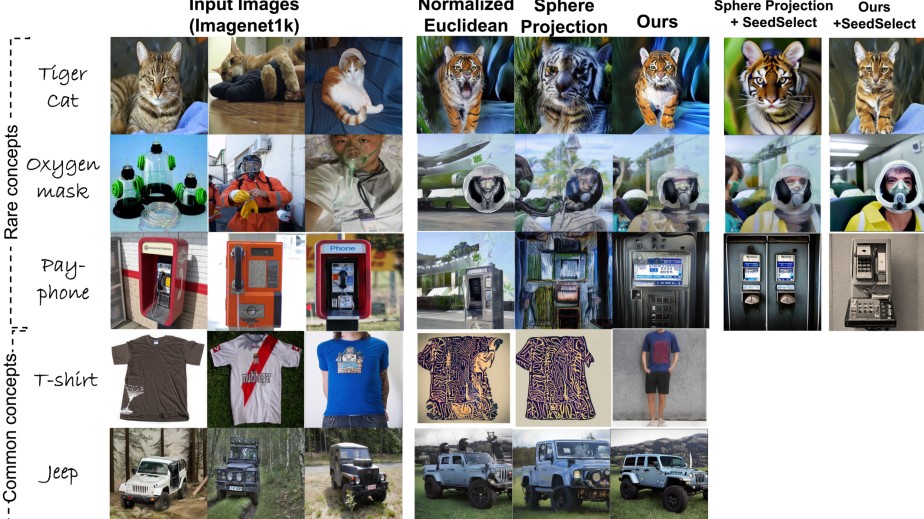

Figure 5: Comparing different centroid optimization approaches on common and rare concepts of ImageNet1k, w.r.t strong baselines. We further initialized SeedSelect [43] with the centroids and run it for up to 5 iterations (∼20 sec on a single A100 GPU). NAO manages to generate correct images for rare concepts (a cat in tiger-cat, a clear oxygen mask and payphone), even without SeedSelect. With SeedSelect the images are further improved. The effect is less emphasized on common concepts w.r.t to strong baselines.

centroid definition boils down to a simple formula - the
average of the seeds. Unfortunately, as we show in Section 3 this definition of a centroid is not suitable for our purposes due to the incompatibility of the centroid's norm with the diffusion model. Instead, we propose to use a generalization of the Euclidean centroid, called the Fréchet mean, which is induced by the distance function $f$ defined above. To achieve this, we formulate an optimization problem that seeks the centroid $c \in \mathbb{R}^d$ that minimizes the distances $f(c, z_T^i)$ to all the seeds:

$$c^*, \gamma_1^*, \ldots, \gamma_k^* = \underset{c, \gamma_1, \ldots, \gamma_k}{\operatorname{argmin}} \left( -\sum_{l=1}^{k} \int_{\gamma_i} \log \mathcal{P}(\gamma_i) ds \right), \tag{3}$$

where $\gamma_i$ are paths between the common centroid $c$ and the inversion $z_T^i$. This optimization problem can be discretized in the same way we discretized Equation (1), i.e. by defining all the paths as piecewise linear paths defined by a sequence of points and adding constraints on the distances between successive points in the path. See the supplemental for a discretized equation.

Figure 4 illustrates an example of a centroid in 2D found with our approach. Figure 5 shows images resulting from the centroids found in seed space. Section 5 provides quantitative results.

**Application to seed optimization methods.** Diffusion models often encounter significant imbalances in the distribution of concepts within their training data [43]. This imbalance presents a challenge for standard pre-trained diffusion models, leading to difficulties in accurately generating rare concepts. One proposed solution, as suggested by [43], involves employing a seed optimization technique. SeedSelect begins with a randomly generated seed that produces an incorrect image and progressively optimizes the seed until a plausible image is generated. However, the method described in [43] is time-consuming, requiring up to 5 minutes to generate a single image. Our approach can serve as an initial point for SeedSelect to reduce substantially its optimization time by generating initial seeds that result in realistic and quality images.

In our case, given few images, generating new data can be obtained in the following manner: First, find a centroid $c^*$ and the interpolation paths $\gamma_i^*$ between image inversion in the seed space. Next, new data is generated by sampling points along the paths from the givens seeds to the centroid, and using them as initializations for SeedSelect.

**Implementation** We implemented a simple optimization algorithm that optimizes the discretized problems using Pytorch and the Adam optimizer. To speed up convergence we initialize the optimization variables: the centroid is initialized with the Euclidean centroid and path variables are initialized as the values of the linear path between the points and the centroid. We implement the constraints $\mathcal{C}(x) = |x_i - x_{i-1}| - \delta \leq 0$ using a soft penalty term in the optimization, by the form $\alpha \cdot \text{ReLU}(\mathcal{C}(x))$ where $\alpha$ is a hyper-parameter. Note that there is a penalty for positive $\mathcal{C}(x)$, when the constraint is not satisfied. We note that in practice there is no guarantee that this optimization scheme converges to the optimal value of the optimization problem, however, we see in practice that high-quality paths are obtained.

## 5 Experiments

We evaluate our approach in terms of image *quality* and also consider generation *time*. We start by studying the quality and semantic content of images generated with our seeding approach and evaluate them in three applications: (1) rare concept generation and (2) semantic data augmentation, to be tested at enhancing few-shot classification, long-tail learning, and few-shot image recognition. (3) We further tested our model on video frame interpolation, showing the results in the supplementary material. An ablation study can also be found in the supplementary material.

**Direct evaluation of interpolation and centroid finding:** Table 1 compares FID scores and the accuracy of images generated using different interpolation and centroid finding methods. For the interpolation experiment, we randomly selected a class from ImageNet1k, and obtained a pair of images and their corresponding seeds through inversion [50]. For interpolation methods that require path optimization, we used paths with 10 sampled points. We then select three seeds along the path (also for LERP and SLERP), with uniform intervals, and feed them into StableDiffusion [38], to generate 3 new images per pair. We repeated the process above to obtain 100 images per class. For the centroid experiment, we used 3-25 seed points obtained from the inversion of additional images (randomly selected) from the train set. We repeated this process for 50 random ImageNet1k classes. Mean FID scores were then calculated (against real ImageNet1k images), along with mean per-class accuracy using a pre-trained classifier. See supplementary material for more details. Our optimized path consistently produces higher-quality images compared to other methods.

We compared our approach to the following baselines: **Euclidean** is the standard Euclidean centroid calculated as the mean of the seeds. **Normalized Euclidean** is the same as Euclidean, but the centroid is projected to the sphere induced by the $\chi$ distribution. **Sphere Projection** first normalizes the seed to a sphere with radius $r = Mode(\chi)$, then finds the centroid on a sphere by optimizing a point that minimizes the sum of geodesic paths between the seeds to the centroid, as presented in [9]. See supplemental for more details. **NAO-path** and **NAO-centroid** are our methods presented in sections 4.1 and 4.2, respectively. The high accuracy and low FID levels of **NAO** in Table 1 demonstrate that our interpolation approach outperforms other baselines in terms of image quality and content. We further put these results to test in downstream tasks (in Sec. 5.1-5.3).

|  | Acc | FID |
|---|---|---|
| **Interpolation methods** | | |
| LERP | 0.0 | 50.59 |
| SLERP [48] | 30.41 | 18.44 |
| **NAO-path (ours)** | 51.59 | **6.78** |
| **Centroid computation methods** | | |
| Euclidean | 0.0 | 54.88 |
| Normalized Euclidean | 27.95 | 37.04 |
| Sphere Projection | 40.81 | 14.28 |
| **NAO-centroid (ours)** | 67.24 | **5.48** |

Table 1: Comparing FID and accuracy of images generated by SD through sampling from different interpolation and centroid computation methods.

### 5.1 Rare-concept generation

Following [43] we compared different centroid optimization strategies in rare-concept generation.

**Dataset.** The evaluation is performed on ImageNet1k classes ordered by their prevalence in the LAION2B dataset [45]. LAION2B [45] is a massive "in the wild" dataset that is used for training foundation diffusion models (*e.g.* Stable Diffusion [38]).

**Compared Methods.** We conducted a comparative evaluation between different centroid estimation strategies. **SeedSelect [43]** is a baseline method where a seed is randomly sampled and no centroid is calculated. Other baselines were presented at the beginning of Section 5.

|  | **ImageNet1k in LAION2B** | | | | | | |
| Methods | **Many** | **Med** | **Few** | **Total Acc** | **FID** | $\hat{T}_{Init}$ | $\hat{T}_{Opt}$ |
|  | n=235 | n=509 | n=256 | | | (sec) | (sec) |
|  | #>1M | 1M>#>10K | 10K># | | | | |
| SeedSelect [43] | 97.8 | 95.8 | 76.1 | 91.3 | 6.5 | 0 | 298 |
| Euclidean | 0.0 | 0.0 | 0.0 | 0.0 | 51.1 | 0 | - |
| Euclidean + SeedSelect | 0.0 | 0.0 | 0.0 | 0.0 | 65.7 | 0 | inf |
| Normalized Euclidean | 45.3 | 39.8 | 20.1 | 36.0 | 45.8 | 0 | - |
| Normalized Euclidean + SeedSelect | 88.1 | 84.8 | 70.9 | 82.0 | 19.2 | 0 | 285 |
| Sphere projection | 52.0 | 48.9 | 26.5 | 44.0 | 12.3 | 0.1 | - |
| Sphere projection + SeedSelect | 97.1 | 95.1 | 74.2 | 90.2 | 8.6 | **0.1** | 72 |
| NAO-centroid (**ours**) | 56.8 | 55.6 | 41.1 | 52.2 | 9.8 | 25 | - |
| NAO-centroid + SeedSelect (**ours**) | **98.5** | **96.9** | **85.1** | **94.3** | **6.4** | 25 | **29** |

Table 2: Image generation quality measured by the accuracy of a pre-trained classifier. Average per-class accuracy is reported separately for the head, tail, and middle classes. NAO produces the best performance in terms of image quality and is also ×10 faster.

| (a) Model | miniImageNet 5-way 5-shot | CIFAR-FS 5-way 5-shot |
| --- | --- | --- |
| Label-Halluc [23] | 67.04 ± 0.7 | 89.37±0.6 |
| FeLMi [39] | 86.08±0.4 | 89.47±0.5 |
| SEGA [59] † | 79.03±0.2 | 86.00±0.2 |
| SVAE [58] ‡ | 80.70±0.2 | 78.89±0.3 |
| Textual Inversion [20]* ‡ | 85.44±3.9 | - |
| Stable Diffusion [38] ‡ | 85.05±0.5 | 90.87±0.5 |
| DiffAlign [40] ‡ | 88.63±0.3 | 91.96±0.5 |
| SeedSelect [43] | 92.08±0.7 | **94.87±0.4** |
| **NAO-centroid (ours)** | **93.21±0.6** | 94.85±0.5 |

| (b) Model | CUB 5-way 5-shot |
| --- | --- |
| TriNet [12] | 84.10±0.4 |
| FEAT [60] | 82.90±0.2 |
| DeepEMD [61] | 88.69±0.5 |
| MultiSem [46] † | 82.9±n/a |
| SEGA [59] † | 90.85±0.2 |
| Stable Diffusion* [38] ‡ | 90.61±0.5 |
| SeedSelect [43] | 96.01±0.4 |
| **NAO-centroid (ours)** | **97.92±0.2** |

| (c) Model | ImageNet-LT |
| --- | --- |
| CE | 41.6 |
| MetaSAug [27] | 47.4 |
| smDragon [42] | 47.4 |
| CB LWS [24] | 47.7 |
| DRO-LT [44] | 53.5 |
| Ride [56] | 55.4 |
| PaCO [13] | 53.5 |
| DRAGON [42] † | 57.0 |
| VL-LTR [52] ! | 70.1 |
| Stable Diffusion [43] ‡ | 56.4 |
| SeedSelect [43]‡ | 74.9±0.5 |
| **NAO-centroid (ours)** | **78.9±0.1** |

Table 3: **(a)-(b) Few-shot recognition**. Comparing NAO to prior work on few-shot learning benchmarks. We report our results with 95% confidence intervals on meta-testing split of the dataset. **(c) Long-tail recognition.** Comparing NAO to prior work on long-tail learning benchmark. Values are accuracy, obtained with a ResNet-50 backbone. * denote results provided by [43]. † for multi-modal methods that use class labels as additional information, ‡ for multi-modal methods that were also pre-trained on external datasets, and ! for multi-modal methods that further finetuned foundation models. Our approach achieves the best results on all benchmarks.

|  | **miniImageNet** | | | **CIFAR-FS** | | | **CUB** | | |
|  | $\bar{T}_{Init}$ | $\bar{T}_{Opt}$ | $\bar{T}_{Total}$ | $\bar{T}_{Init}$ | $\bar{T}_{Opt}$ | $\bar{T}_{Total}$ | $\bar{T}_{Init}$ | $\bar{T}_{Opt}$ | $\bar{T}_{Total}$ |
| --- | --- | --- | --- | --- | --- | --- | --- | --- | --- |
| SeedSelect [43] | - | 276 | 276 sec | - | 228 sec | 228 sec | - | 312 sec | 312 sec |
| **NAO-centroid (ours)** | 28 sec | 25 sec | **53 sec** | 29 sec | 21 sec | **50 sec** | 31 sec | 29 sec | **60 sec** |

Table 4: Comparing convergence time between SeedSelect [43] alone and SeedSelect initialized with seeds found using NAO. Note the five-fold reduction in runtime for our method.

**Experimental Protocol.** For every class in ImageNet, we randomly sampled subsets of training images, calculated their centroid in seed space, and generated an image using Stable Diffusion directly or as input to SeedSelect [43]. The class label was used as the prompt. This process is repeated until 100 images are generated for each class. We then used a SoTA pre-trained classifier [53] to test if the generated images are from the correct class or not (more details can be found in the supplementary). We use this measure to evaluate the quality of the generated image, verifying that a strong classifier correctly identifies the generated image class. We also report Mean FID score between the generated images and the real images, mean centroid initialization time $\hat{T}_{Init}$ and mean SeedSelect optimization time until convergence $T_{Opt}$ on a single NVIDIA A100 GPU. The results summarized in Table 2 show that our NAO method substantially outperforms other baselines, both in accuracy and in FID score. Furthermore, NAO gives a better initialization point to SeedSelect [43], yielding significantly faster convergence without sacrificing accuracy or image quality.

Next, we evaluate NAO as a semantic data augmentation method on two learning setups: (1) Few-shot classification, and (2) Long-tail classification. We aim to show that our approach not only achieves faster generation speed but also attains state-of-the-art accuracy results on these benchmarks.

## 5.2 Few-shot learning

Few-shot benchmarks typically provide a limited number of samples per class, along with the corresponding class labels. NAO is used as a semantic data augmentation approach: We use the given samples and generate many more samples from that class, which are then used to train a classifier.

**Datasets.** We evaluated NAO on three common few-shot classification benchmarks: **(1) CUB-200 [55]:** A *fine-grained* dataset comprising 11,788 images of 200 bird species. The classes are divided into three sets, with 100 for meta-training, and 50 each for meta-validation and meta-testing. **(2) miniImageNet [54]:** A modified version of the standard ImageNet dataset [14]. It contains a total of 100 classes, with 64 classes used for meta-training, 16 classes for meta-validation, and 20 classes for meta-testing. The dataset includes 50,000 training images and 10,000 testing images, with an equal number of images distributed across all classes. **(3) CIFAR-FS [8]:** Created from CIFAR-100 [26] by using the sampling criteria as miniImageNet. Has 64 classes for meta-training, 16 classes for meta-validation, and 20 classes for meta-testing; each class containing 600 images.

Following all previous baselines, we report classification accuracy as the metric. We report our results with 95% confidence intervals on the meta-testing split of the dataset.

**Compared Methods.** We conducted a comparative evaluation of our approach with several state-of-the-art methods for few-shot learning. These methods fall into three categories based on their approach. (A) Methods that do not use pre-training nor use class labels during training: **Label-Hallucination [23]** and **FeLMi [39]**; (B) Methods that use class labels during training: **SEGA [59]**; and (C) Methods that utilize a classifier pre-trained on external data and also use class labels during training: **SVAE [58]**, **Vanilla Stable Diffusion (Version 2.1) [38]**, **Textual Inversion [20]**, **DiffAlign [40]**, and **SeedSelect [43]**. The last four methods are semantic data augmentation methods.

**Experimental Protocol.** For a fair comparison with prior work, we follow the training protocol in [40] and [43]. We generated 1,000 additional samples for each novel class using SeedSelect. It was initialized with seeds found with NAO using the centroid and interpolation samples of the few-shot images provided during meta-testing and prompted it with the corresponding class name. We used a ResNet-12 model for performing N-way classification and trained it using cross-entropy loss on both real and synthetic data.

**Results.** Tables 3a and 3b compare NAO with SoTA approaches on few-shot classification benchmarks: CUB, miniImageNet, and CIFAR-FS. NAO consistently outperforms all few-shot methods on CUB [55] and miniImageNet [54], and reaches comparable results to SeedSelect [43] on CIFAR-FS [8]. Table 4 further compares the mean run time of SeedSelect with and without NAO on these datasets, on a single NVIDIA A100 GPU. The results highlight the competence of our approach in generating rare and fine-grained classes, to reach top accuracy with a five-fold reduction in the runtime.

## 5.3 Long-tail learning

**Datasets.** We further evaluated NAO on long-tailed recognition task using the **ImageNet-LT [32]** benchmark. ImageNet-LT [32] is a variant of the ImageNet dataset [15] that is long-tailed. It was created by sampling a subset of the original dataset using the Pareto distribution with a power value of $\alpha = 6$. The dataset contains 115,800 images from 1,000 categories, with the number of images per class ranging from 5 to 1,280.

**Compared Methods.** We compared our approach with several state-of-the-art long-tail recognition methods. These methods fall into three categories based on their approach. (A) Long-tail learning methods that do not use any pretraining nor employ class labels for training: **CE** (naive training with cross-entropy loss), **MetaSAug [27]**, **smDragon [42]**, **CB LWS [24]**, **DRO-LT [44]**, **Ride [56]** and **Paco [13]**. (B) Methods that use class labels as additional information during training: **DRAGON [42]**. (C) Methods that were pre-trained on external datasets and use class labels as additional information during training: **VL-LTR [52]**, **Vanilla Stable Diffusion (Version 2.1) [38]** and **SeedSelect [43]**. **MetaSAug [27]**, **Vanilla Stable Diffusion** and **SeedSelect [43]** are semantic augmentations methods. Note that **VL-LTR [52]**, compared to other models, further fine-tuned the pre-trained model (CLIP [35]) on the training sets.

**Experimental Protocol.** Following previous methods, we use a ResNet-50 model architecture, train it on real and generated data, and report the top-1 accuracy over all classes on class-balanced test sets.

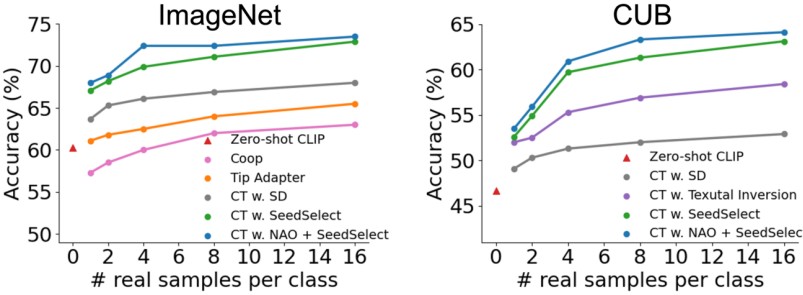

Figure 6: Results for few-shot image recognition, comparing NAO+SeedSelect to previous approaches. Fine-tuning a CLIP classifier on SeedSelect generated images initialized with NAO consistently achieves SOTA results across all shot levels.

**Results.** Tables 3c evaluates our approach compared to long-tail recognition benchmarks. NAO reaches SoTA results albeit simple, compared to other complex baselines.

### 5.4 CLIP finetuning with Synthetic data for image classification

We follow [43, 62, 64] and further examine the advantage of using our method in the generation of synthetic images, as semantic augmentation, for the downstream task of few shot CLIP recognition. In order to deal with rare concepts we use our method together with SeedSelect showing SoTA results in this task [43]. To this end we use NAO as seed initialization for SeedSelect. In the context of this task, we are provided with a limited number of real training samples per class, along with their corresponding class names, and the goal is to fine-tune CLIP.

**Experimental Setup.** For a fair comparison we follow the same experimental protocol of [43]. Specifically, we generated 800 samples for each class using NAO+SeedSelect from the limited real training samples. We then fine-tune a pre-trained CLIP-RN50 (ResNet-50) through mix-training, incorporating both real and generated images.

**Compared Methods.** We compare our approach with the following baselines: **Zeor-shot CLIP:** Applying the pre-trained CLIP classifier without fine-tuning; **CooP [64]:** Fine-tuning a pre-trained CLIP via learnable continuous tokens while keeping all model parameters fixed; **Tip Adapter [62]:** Fine-tuning a lightweight residual feature adapter; **CT & SD:** Classifier tuning with images generated with SD. **Textual Inversion [20]:** Classifier tuning with images generated using personalized concepts (See Supp for implementation detail). Results for **CT & SD** were reproduced by us on SDv2.1 using the code published by the respective authors.

**Results.** Figure 6 shows results for the few-shot image recognition task. It demonstrates that initiating seeds using NAO improves SeedSelect performance on all shots.

## 6   Conclusion

This paper proposes a set of simple and efficient tools for exploring the seed space of text-to-image diffusion models. By recognizing the role of the seed norm in determining image quality based on the $\chi$ distribution as prior, we introduce a novel method for seed interpolation and define a non-Euclidean metric structure over the seed space. Furthermore, we redefine the concept of a centroid for a set of seeds and present an optimization scheme based on the new distance function. Experimental results demonstrate that these optimization schemes, biased toward the $\chi$ distribution mode, generate higher-quality images compared to other approaches. Despite the simplicity and effectiveness of our approach, there are several limitations to be aware of. Firstly, compared to standard interpolation and centroid calculation, it involves an additional optimization step. Secondly, our centroid and/or the samples along our interpolation paths may not produce plausible and semantically correct images on their own, necessitating the use of SeedSelect optimization. Lastly, although our method is expected to be applicable to all diffusion models, we specifically evaluated it with the open-source Stable Diffusion [38] model in this study.

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
