# Supplemental Material

## A   Proof for proposition 1

**Proposition 1.** *Let $W : \mathbb{R}^d \to \mathbb{R}$ be a strictly positive continuous function. Given $x, y \in \mathbb{R}^d$, define $f(x, y)$ as the infimum over all path integrals from $x$ to $y$ where the infimum is taken over all piecewise differentiable curves.*

$$f(x, y) := \inf_{\gamma} \int_{\gamma} W(\gamma) ds \qquad s.t. \quad \gamma(0) = x, \gamma(1) = y. \tag{4}$$

*Then $f$ is a distance function*

*Proof.* We need to show that $f$ satisfies three properties: non-negativity, symmetry, and the triangle inequality.

First, $f(x, y) \geq 0$ for all $x, y \in \mathbb{R}^d$. This follows from the fact that any line integral of a positive function is non-negative, and from the fact that the infimum of a set of non-negative numbers is itself non-negative. In addition, we have to show that $f(x, y) = 0$ iff $x = y$. If $x = y$ then $W$ has zero line integral on the constant curve $\gamma \equiv x$ which implies $f(x, y) \leq 0$, which combined with the non-negativity of $f$ implies $f(x, y) = 0$.

In the other direction assume that $x \neq y$ so $\|x - y\| \geq 0$. We will show that for any such $x, y$ there is a constant $c > 0$ such that the line integral over $\gamma$ is strictly larger than $c$ which implies $f(x, y) \geq c > 0$. Let $K$ be a closed ball that contains $x, y$ in its interior. Let $\epsilon > 0$ be the minimal distance between $x$ and $\partial K$. let $m_K > 0$ be the minimal value of $W$ on $K$ that exists due to compactness of $K$, and the positivity and continuity of $W$. finally, Let $\gamma$ be a path from $x$ to $y$. We split into two cases:

1. if $Im(\gamma) \subset K$ then we have $f(x, y) \geq m_k \|x - y\| > 0$ since the value of any line integral from $x$ to $y$ is bounded by the distance between the points times the minimal value of the integrand.

2. Otherwise, $\gamma$ has to intersect with $\partial K$ at some point $x'$ and a similar argument shows that $f(x, y) \geq m_k \|x - x'\| \geq m_k \epsilon > 0$.

This implies that for any $\gamma$ the value of the line integral is larger than $c = \min(m_k \epsilon, m_k \|x - y\|) > 0$ and concludes this part of the proof.

Next, it is easy to see that $f(x, y) = f(y, x)$ for all $x, y \in \mathbb{R}^d$. This follows from the fact that the reverse path gives rise to the same line integral value as the original path.

Finally, we show that $f(x, z) \leq f(x, y) + f(y, z)$ for all $x, y, z \in \mathbb{R}^d$. Let $\epsilon > 0$. Following the infimum definition there is a path $\gamma_1$ from $x$ to $y$ such that $\int_{\gamma_1} W(\gamma_1) ds < f(x, y) + \epsilon/2$, and a path $\gamma_2$ from $y$ to $z$ such that $\int_{\gamma_2} W(\gamma_2) ds < f(y, z) + \epsilon/2$. Then, the path $\gamma$ obtained by concatenating $\gamma_1$ and $\gamma_2$ is a piecewise differentiable curve from $x$ to $z$. Therefore,

$$\begin{aligned} f(x, z) &\leq \int_{\gamma} W(\gamma) ds \\ &= \int_{\gamma_1} W(\gamma_1) ds + \int_{\gamma_2} W(\gamma_2) ds \\ &< f(x, y) + f(y, z) + \epsilon. \end{aligned}$$

Since this is true for any $\epsilon > 0$ we get $f(x, z) \leq f(x, y) + f(y, z)$ as needed. Thus, we have shown that $f$ satisfies all three properties of a distance function, and therefore $f$ is a distance function. $\square$

To prove Proposition 1 we use the lemma above and the fact that the $\log(\chi^d(x))$ is positive for all $x \in \mathbb{R}^d$.

## B  Background and preliminaries

**Diffusion models:** Text-guided diffusion models aim to map a random noise $z_t$ and textual condition $P$ to an output image $z_0$, which corresponds to the given conditioning prompt. The mapping between $z_t$ to $z_0$ using $P$, also called the "denoising process", is performed sequentially by a network $\varepsilon_\theta$ which is trained to predict noise by minimizing the loss: $\mathcal{L} = \mathbb{E}_{z_0,y,\varepsilon \sim \mathcal{N}(0,1),t}\left[||\varepsilon - \varepsilon_\theta(z_t, t, c(y))||_2^2\right]$. Here, $z_t$ is a noised sample, and $c(P)$ is the embedding of the text condition, where noise is added to the sampled data $z_0$ according to a timestamp $t$. Essentially, the role of the denoising network $\epsilon_\theta$ is to accurately eliminate the added noise $\epsilon$ from the latent code $z$ at every time step $t$, based on the input of the noisy latent code $z_t$ and the encoding of the conditioning $c(P)$. At inference, $z_T$ is sampled from a standard multivariate Gaussian distribution $\mathcal{N}(0, I)$ and the noise is iteratively removed by the trained $\epsilon_\theta$ for T steps, resulting $z_0$.

Although our method is universally applicable to all diffusion models, in this study we specifically employed the open-source Stable Diffusion [46]. Here, the diffusion process is applied on a latent image encoding $z_0 = \mathcal{E}(x_0)$ and an image decoder is employed at the end of the diffusion backward process $x_0 = \mathcal{D}(z_0)$.

**Sampling and Inversion:** The process of mapping an image to noise is a Markov chain starting from $z_0$, and gradually adding noise to obtain latent variables $z_1, z_2, \ldots, z_T$. The sequence of latent variables follows $q(z_1, z_2, \ldots, z_T) = \Pi_{i=1}^t q(z_t|z_{t-1})$. A step in this process is a Gaussian transition $q(z_t|z_{t-1}) := \mathcal{N}(z_t, \sqrt{1-\beta_t} z_{t-1}, \beta_t I)$ parameterized by a schedule $\beta_0, \beta_1, \ldots, \beta_T \in (0,1)$. Note that $z_t$ can be expressed as a linear combination of noise and $z_0$: $z_t = \sqrt{\alpha_t z_0} + \sqrt{1 - \alpha_t} w$, where $w \sim \mathcal{N}(0, I)$ and $\alpha_t = \Pi_{i=1}^t (1 - \beta_i)$.

Reversing the process is not immediately obvious and thus several schedulers were proposed [23, 26, 31, 58]. In this paper, we employ DDIM [58] scheduler, a popular deterministic scheduler. Other deterministic scheduler would be suitable, and we show in section I below that our method performs well with other schedulers. For DDIM, [58] proposed denoising in the following way:
$z_{t-1} = \sqrt{\frac{\alpha_{t-1}}{\alpha_t}} z_t + (\sqrt{\frac{1}{\alpha_{t-1}} - 1} - \sqrt{\frac{1}{\alpha_t} - 1}) \cdot \varepsilon_\theta(z_t, t, c(y))$.

Since DDIM is deterministic, its inversion can be obtained based on the assumption that the Ordinary Differential Equation (ODE) process can be reversed in the limit of small steps. $z_{t+1} = \sqrt{\frac{\alpha_{t+1}}{\alpha_t}} z_t + (\sqrt{\frac{1}{\alpha_{t+1}} - 1} - \sqrt{\frac{1}{\alpha_t} - 1}) \cdot \varepsilon_\theta(z_t, t, c(y))$. See [19, 58] for more details. Using this technique we can inverse a real image $z_0$ to its latent $z_t$ in the seed space.

## C  Analysis of Approximation and Optimization Errors

In this section, we delve into the approximation and optimization aspects of our proposed methodology, particularly with regard to Equation (1) and Equation (2). Solving the optimization problem stated in Equation (2) yields an approximation to its continuous counterpart, Equation (1). There are two primary error components:

(i) Approximation Error: This error arises from the discreteness of the integral in Equation (2) as compared to the continuous integral in Equation (1). However, we assert that due to the smoothness of the function f(x), the minimizer of Equation (1) is expected to exhibit smooth behavior. Consequently, we can use piecewise linear paths in Equation (2) to approximate it with arbitrarily small error by increasing the number of points.

(ii) Optimization Error: This error stems from the optimization process applied to Equation (2). In line with common practice in deep learning, we employ first-order optimization techniques to optimize a non-smooth function. We show that our experimental results and 2D visualizations support the assertion that our optimization procedure converges to a satisfactory solution.

From a practical standpoint, despite the potential lack of numerical exactness, our approach consistently yields high-quality results, as indicated by the Frechet Inception Distance (FID) measures, which serve as a metric for evaluating the quality of generated images.

## D   Discretizing the centroid-finding problem

To approximate the solution to problem Eq. 3 in practice, we discretize the paths to the centroid in a similar fashion to Equation (5) by representing them as a sequence of piece-wise linear segments. We then replace the integral with its corresponding Riemann sum over these piece-wise linear paths:

$$\underset{c, x_0^l, \ldots, x_n^l \, \forall l}{\text{minimize}} \quad -\log \mathcal{P}(c) - \sum_{l=1}^{k} \sum_{i=1}^{n} \log \mathcal{P}\left(\frac{x_i^l + x_{i-1}^l}{2}\right) \|x_i^l - x_{i-1}^l\| \tag{5}$$

$$\text{s.t.} \quad x_0^l = c, x_n^l = z_T^l, \quad \|x_i^l - x_{i-1}^l\| \leq \delta, l \in \{1, \ldots, k\}, i \in \{1, \ldots, n\}$$

where $i$ indexes the points along a path between the centroid and an inversion point, and $l$ indexes the *paths*. The minimization drives the optimization toward a centroid $c$ that has a minimum overall distance to all reference points in terms of our new metrics.

## E   FID and per-class accuracy with a pre-trained classifier

Figures 2(b)-2(c) in the main paper provide FID and per-class accuracy of a pre-trained classifier for images generated by Stable Diffusion. Tables 2,1 further evaluate different interpolation and centroid evaluations with these metrics. Here we provide more details about these experiments.

FID (Fréchet inception distance) [25, 59] was calculated between generated images (typically 100 images per class) and real ImageNet1k test images (50 images for each class). To calculate the FID score, we used the features of a pre-trained ImageNet1k classifier. Specifically, we used InceptionV3 with 64-dim feature vectors.

For per-class accuracy, we follow [51] and used a SoTA pre-trained ImageNet classifier. Specifically, we used MaxViT image classification model [62] pre-trained on ImageNet-21k (21843 Google specific instance of ImageNet-22k) and fine-tuned on ImageNet-1k. It achieves a top-1 of 88.53% accuracy and 98.64 top-5 accuracy on the test-set of ImageNet. This classifier measures the correctness of generated images.

## F   A *Spherical Projection* baseline

The "Spherical Projection" baseline [9] in the main paper tries to find a centroid on a sphere. Specifically, it optimizes a point that minimizes the sum of geodesic distances between all input points (inversions). This can be formulated as follows. Given $z_T^1, z_T^2, \ldots, z_T^k$ points (projected) on a sphere with radius $R$. We would like to find a point $c$ such that

$$c^* = \underset{c}{\text{argmin}} R \sum_{i=1}^{k} \arccos(c \cdot z_T^i) \quad , \tag{6}$$

where $R \cdot \arccos(c \cdot z_T^i)$ the "arc length" and the geodesic distance between the centroid $c$ and another point $z_T^i$. These extrema can be solved using Lagrangian multipliers, with $c$ on the sphere, as the constraint: $L(c, \lambda) = \sum_{i=1}^{k} R \cdot \arccos\left(c \cdot z_T^i\right) + \lambda(R - c \cdot c)$. Taking partial derivatives results with:

$$\frac{\partial L}{\partial C_j}(c, \lambda) = -R \sum_{i=1}^{k} \frac{z_T^i(j)}{\sqrt{1 - (c \cdot z_T^i)^2}} - 2\lambda c_j \tag{7}$$

$$\frac{\partial L}{\partial \lambda}(c, \lambda) = R^2 - c \cdot c \quad , \tag{8}$$

where $z_T^i(j)$ is the $j-$th element of $z_T^i$. The extrema is then:

$$c = m \sum_{i=1}^{k} \frac{z_T^i}{\sqrt{1 - (c \cdot z_T^i)^2}}, \tag{9}$$

where $m$ a constant selected such that the second normalization constraint holds. An iterative equation was obtained, and the centroid is determined through iterative processes until convergence is achieved.

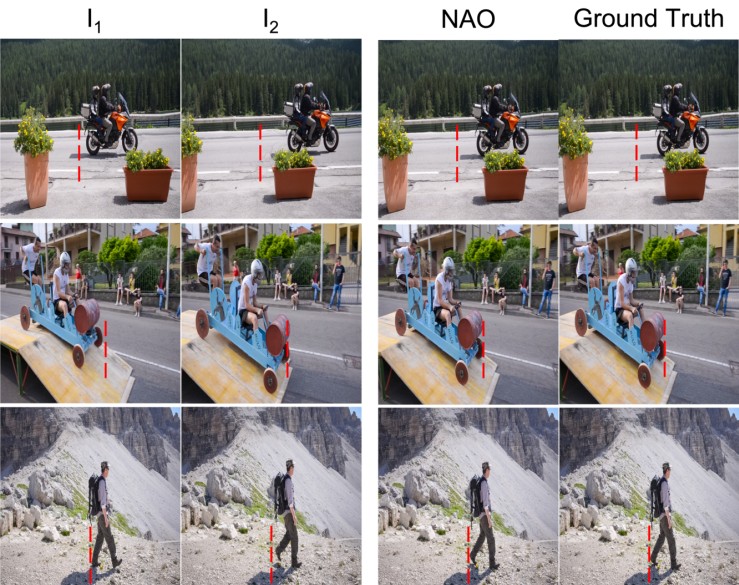

| | I₁ | I₂ | NAO | Ground Truth |

Fig. S 1: In the video frame interpolation task, NAO can produce intermediate frames through interpolating between seed inversions of two provided frames (I1 and I2). The red overlaid lines are at fixed locations and were added for sake of visualization. The generated intermediate frames from seed interpolation are highly close to the ground truth, and contain the subtle temporal changes.

| | PSNR
(Higher is better) | SSIM
(Lower is better) |
|---|---|---|
| **MCVD** [65] | 18.646 | 0.705 |
| **LDMVFI** [14] | 25.541 | 0.833 |
| **NAO-path (ours)** | 25.413 | 0.813 |

Table S 1: Results of NAO for video frame interpolation task on the DAVIS [41] dataset. NAO achieves comparable results to existing methods.

## G   Video frame interpolation

We further applied our approach to the task of video frame interpolation, aiming to generate an intermediate frame between existing consecutive frames (I1 and I2) in a video sequence. Using NAO, we interpolated between the seed inversions of two given frame images, optimizing 50 points, and generating an image from the middle points. Example images in Figure S1 illustrate NAO's ability to create intermediate frames. To further demonstrate NAO's capability, we conducted also a quantitative evaluation. To this end, we followed the experiment outlined in [14, 65], focusing on the DAVIS dataset [41]: a widely acknowledged benchmark for Video Frame interpolation tasks. The evaluation of predicted frames against the ground truth was carried out using established metrics like PSNR and SSIM. Table S1 demonstrates that our interpolation approach achieves comparable results to existing methods specifically tailored for video interpolation, despite solely using a pre-trained text-to-image model.

## H   Experiments on few-shot and long-tail learning

We provide here additional information and implementation details regarding experiments on Few-shot and Long-tail learning.

**Few-shot learning:**   We followed the training protocol of [48] and evaluated NAO for 600 episodes. At each episode, base and novel classes are randomly selected. We generated 1000 additional samples for each novel class using NAO and SeedSelect [51]. Specifically, SeedSelect was initialized with centroids and samples from the interpolation paths found by NAO using the 5-shot images provided

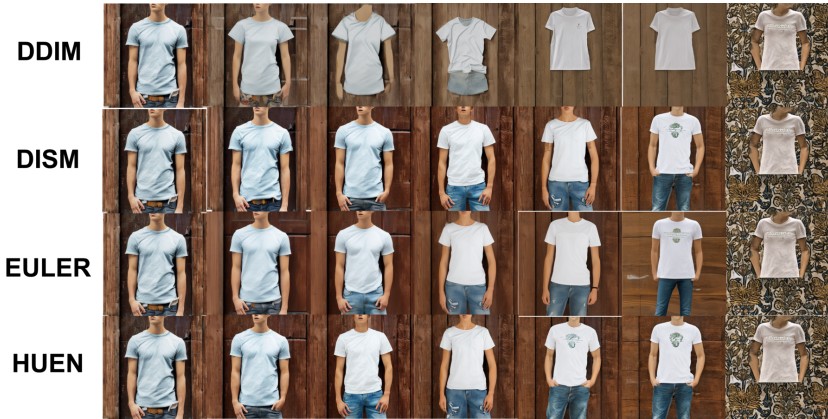

Fig. S 2: Images generated by different schedulers on the same interpolation path between two seeds optimized by NAO. Images were generated with 20 denoising steps. Our approach results in high-quality images regardless of the scheduler used in the method.

during meta-testing. The mean accuracy is calculated for each episode. Finally, we reported the mean accuracies along all episodes with 95% confidence intervals.

For NAO + SeedSelect to generate 1000 total images for each class, we optimized paths with a length of 200 points between the centroid of the 5-shot training samples and the samples themselves.

To ensure a fair comparison, we employed the ResNet-12 architecture as the backbone network. On top of the feature extractor, we incorporate a two-layer MLP for N-way classification. A stochastic gradient descent (SGD) optimizer is used with a momentum of 0.9. The learning rates for the backbone and classifier are respectively set to 0.025 and 0.05, accompanied by a weight decay of 5e-4. During training, standard data augmentations such as color jittering, random crop, and horizontal flips are applied within a minibatch consisting of 250 images.

**Long-tail learning:**  NAO was further evaluated as a semantic augmentation approach on the ImageNet-LT [38] benchmark. Here, we generated samples for each class until the combined total of real and generated samples equaled the count of the class with the highest number of samples in the dataset, resulting in a uniform data distribution. Similarly to the approach used for few-shot benchmarks, we generated images for each class by initializing SeedSelect with centroids and samples obtained from interpolation paths discovered between training samples to their centroid. We optimized paths ranging in length from 10 to 200, adjusting based on the available number of samples for each class. We measure the accuracy for each class and report the mean for all classes.

For a fair comparison, we follow prior work [29, 38, 50] and train a ResNet-50 backbone. The backbone was trained on real and generated data using a stochastic gradient descent (SGD) optimizer with a momentum of 0.9. Additionally, we used the cosine learning-rate schedule to train the network. During training, standard data augmentations such as color jittering, random crop, and horizontal flips are applied within a minibatch consisting of 250 images.

**Hyper-parameter tuning:**  We determined the number of training epochs (early-stopping), and the learning rate of the backbone network using the validation set provided.

# I   Sensitivity to scheduler

In the main paper, we used DDIM scheduler both for the diffusion forward process (seed to image) and inversion process (image to seed). Here we show that our method can be also used with another deterministic scheduler. Figure S2 shows images generated by sampling from NAO-path using DDIM [58], DEIS [73] Determenistic Euler Scheduler [31] and Heun [31] schedulers. Note that while Euler and Heun schedulers rescale the input latent, they still initiate their process with a sample from a multi-variant Gaussian distribution. This characteristic allows our approach to be potentially applicable to all schedulers.

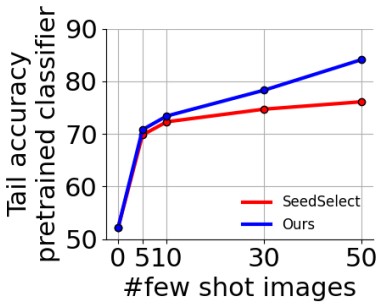

Fig. S 3: Effect of number of few-shot samples on accuracy of a pre-trained ImageNet-1k classifier.

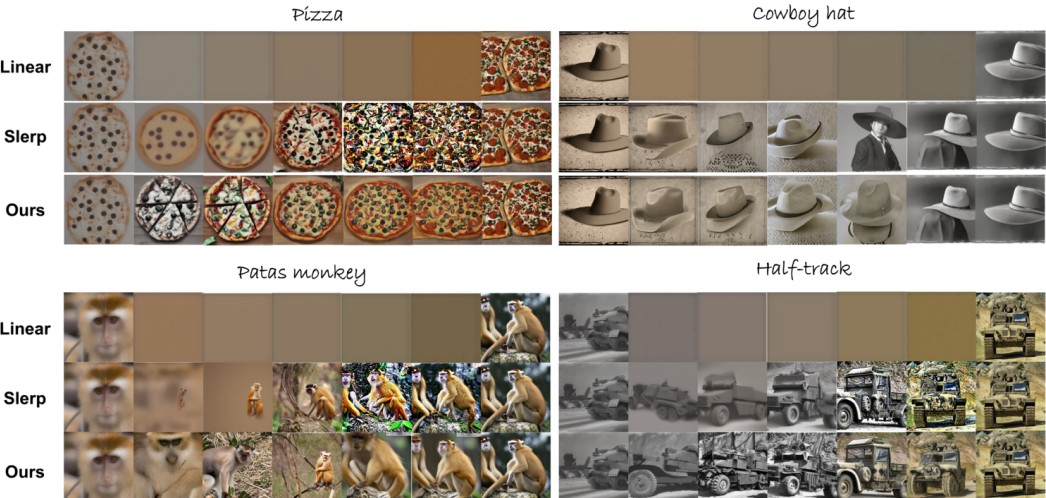

Fig. S 4: Additional qualitative comparison for different interpolation methods between two image seeds.

## J   Ablation study

### J.1   Dependence on the number of samples ("shots")

Figure S3 illustrates the comparison between our suggested method NAO and SeedSelect [51] regarding the impact of varying the number of training samples (#shots) on generation quality. The evaluation was performed on tail classes of ImageNet1k [51]. We tested per-class tail accuracy for changing the number of training samples using a pre-trained ImageNet1k classifier (see Section E). The increase in the number of training samples leads to the generation of more semantically correct images and it is evident that NAO achieves superior accuracy results compared to SeedSelect [51].

### J.2   Diversity

We follow [51] and analyze the diversity of images generated by NAO. To analyze the diversity of our generated images, we used two metrics: The number of statistically different beans (NDB) [45] and the entropy across clusters. While NDB tries to locate mode collapse, the entropy metric makes sure that the generated images are diverse.

Specifically, we divided the test set of each ImageNet class into 50 clusters (one for each test sample). Subsequently, we allocated the images generated by NAO to these clusters and evaluated the NDB and entropy within the clusters. On average, the entropy across all classes was 3.9 bits, whereas the test set clusters had an entropy of 5 bits. Furthermore, the NDB value was 2.21. These findings indicate

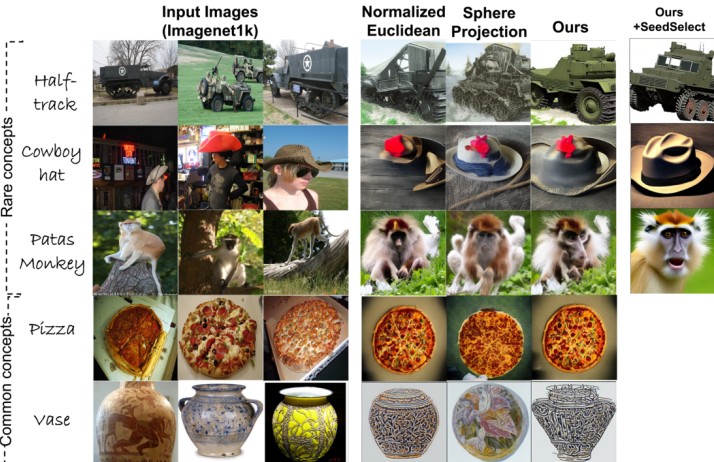

Fig. S 5: Additional qualitative comparison for different centroid optimization approaches on common and rare concepts of ImageNet1k. We further initialized SeedSelect [51] with the centroids and run it for up to 3 iterations ($\sim$15 sec on a single A100 GPU). While alternative methods demonstrate satisfactory performance with common concepts, they frequently lack visual credibility when employed with uncommon objects.

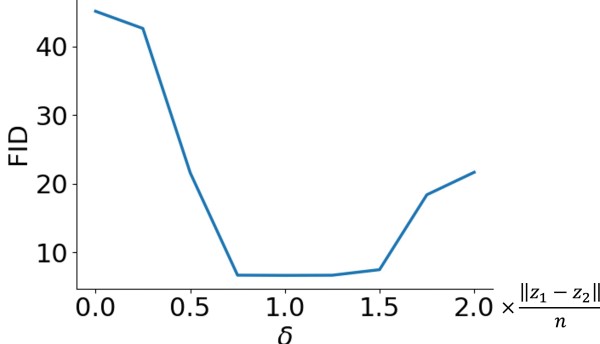

Fig. S 6: Analysis examining the influence of various $\delta$ values on the FID of images generated along the interpolation trajectory. The investigation demonstrates that extremely low or high delta values detrimentally impact FID scores. Such values either result in excessive proximity between points, causing overlap or overly disperse them, steering the path into regions of low likelihood.

that the generated images possess a level of diversity comparable to real images, demonstrating a lack of mode collapse.

We acknowledge that there are potential alternative methods for quantifying diversity [63]. The exploration of diversity is still in its early stages and has not been thoroughly examined. However, this area holds significant potential for future research and investigation, offering valuable opportunities for further exploration.

### J.3 $\delta$ analysis

In the experiments done in the main paper, we set $\delta$ to be $||z_1 - z_2||/n$, where $z_1$ and $z_2$ are seed inversions of real images ($x_1$ and $x_2$), and $n$ is the number of interpolation points to be optimized. We further conducted an in-depth analysis to investigate the impact of different delta values on the FID of images generated along the interpolation path. Figure S6 presents the results of this analysis. The findings reveal that excessively low or high delta values adversely affect the FID, as they either

constrain the points too closely together, causing overlap, or spread them too far apart, leading the path into low-likelihood regions.

# K   Additional qualitative results

Figure S4 provides qualitative analysis and compares NAO to LERP and SLERP [56]. Figure S5 further provides a qualitative comparison between NAO and other centroid finding approaches. It is evident from the figures that our approach samples better paths resulting in higher quality images, particularly in rare concepts.