# OpenReview forum: "Norm-guided latent space exploration for text-to-image generation"
_NeurIPS.cc/2023/Conference — NeurIPS 2023 poster_

### Official Review · Reviewer_yJ89 · 2023-07-05

**Soundness:** 3 good
**Presentation:** 3 good
**Contribution:** 3 good
**Rating:** 6
**Confidence:** 4

**Summary:**

	This paper proposes a novel method for interpolating between two seeds and demonstrates that it defines a new non-Euclidean metric that takes into account a norm-based prior on seeds. This paper describes a simple yet efficient algorithm for approximating this metric and using it to further define centroids in the latent seed space, which helps generate rare concept images and leads to state-of-the-art performance on few-shot and long-tail benchmarks.

**Strengths:**

-	This paper first discusses the property of the seed, that is, the relationship between the norm of the seed and the quality of the generated image, which provides good theoretical support for the proposed new non-Euclidean metric.
-	The newly proposed non-Euclidean metric combined with the centroid method has a good effect according to the experimental results and has been optimized to a certain extent for problems such as rare concept generation and long tail training.
-	From the seed level, the paper investigates the text-image generation problem of the diffusion model, and verify the feasibility of controlling the image generation from the seed level.


**Weaknesses:**

-	In the early stage, the norm problem of seed was elaborated and verified too much, but the core path optimization and centroid method did not elaborate enough.
-	Judging from the experimental results of the pictures in the article, the method in this paper does not seem to show a particularly great advantage, especially after adding the seed-selection method.
-	The two interpolation methods compared in this article are very basic. There are more non-linear interpolation methods to compare, and the results after interpolation seem to be inconsistent with the actual results. In my understanding (also my experiment testing), no matter what kind of Gaussian noise the seed is, SDM can generate a relatively reasonable image, rather than an unnatural noisy image like the ones presented in Figure 1(left).
-	From the point of view of experimental design, the seed select method is also a key part, such as Figure 5, but it has not been explained in detail.


**Questions:**

-	There are some formatting errors in the article, such as the picture in the upper right corner of page 4 without icons and annotations.
-	I can not understand the interpolation path of 2D space. Hope to have more explanations of 2D space, and how to get the result such as in Figure 1 (right).
-	Other concerns have already been mentioned in Weakness.


**Limitations:**

Yes, the authors have addressed the limitations and potential negative societal impact of their work.

---

> ### Author Rebuttal · Authors · 2023-08-09
>
> Thank you for finding our approach effective with good theoretical support. We address your comments below.
>
> #### **Q1: Core path optimization and centroid method did not elaborate enough.**
> **A1:** We value the reviewer’s feedback to improve our paper. Due to lack of space, we provide more details on this in Sec. C in the Supplementary material. We will elaborate on this section in the final version with additional details.
>
> #### **Q2: The method lacks an advantage with seed-selection, based on results.**
> **A2:** Please refer to the shared response (A1) provided in the shared answer for all reviewers. NAO has two main advantages over SeedSelect: It is 10x faster, and it improves quality as measured by an improvement in terms of FID and accuracy.  See Table2 for the numbers.
>
> #### **Q3: The two interpolation methods compared in this article are very basic**
> **A3:** We compared to SoTA approaches in latent space interpolation, and agree that smarter methods can be used. However, we argue that being aware of the latent space structure plays a significant role, much more than using smarter interpolation methods. The key factor that was so far neglected in latent space interpolation is being aware of the distribution of samples in that space. This is the main contribution of this paper. We are open to evaluating novel methods upon request and appreciate the reviewer's observation, which we acknowledge as a potential avenue for future exploration.
>
> #### **Q4: No matter what kind of Gaussian noise the seed is, SD can generate a relatively reasonable image.**
> **A4:** Thank you for your comment, however, we would like to clarify that this may not be entirely accurate. SD generates reasonable images when the norm of the seeds is close to the mode of the chi-distribution (L 115-123 in the main paper). Randomly sampling from a Gaussian distribution and applying current interpolation techniques results, most of the time, with visually appealing images because the seed norms possess this property. However, in practical scenarios with **real images**, the properties of seeds obtained from inversions may not always be favorable, leading to the failure of current interpolation methods. Our approach considers this prior knowledge and selects seeds that allow SD to generate plausible results.
>
> #### **Q5: [1] was not been explained in detail.**
> **A5:** Indeed, [1] plays a crucial role in rare-concept generation. However, it is important to note that our approach does not depend on SeedSelect and operates independently. Nevertheless, in the final version, we will provide a more elaborate explanation of [1] and offer additional details about their methodology.
>
> #### **Q6: 2D space examples and Figure 1 are difficult to understand.**
> **A6:** We appreciate the reviewer's valuable feedback and are grateful for their efforts to improve our paper. In these figures, the colors correspond to the log-likelihood of the chi distribution. The black points represent real images, while the colored points represent interpolation paths/centroids between these real images. Our approach prioritizes points with high log-likelihood, leading to visually appealing results when using SD. Conversely, other methods disregard the inherent structure of the seed space, resulting in poor-quality images. We acknowledge the need for additional clarification on these figures and will provide more detailed explanations in the text for the final version of the paper and the camera-ready version.
>
> #### **Q7: Formatting errors and typos.**
> **A7:**  Thank you. Will be fixed in the camera-ready version.
>
> #### **References**
> [1].   Samuel et al. (2023), "It's all about where you start: text to image generation with seed selection"

---

> > ### Comment · Reviewer_yJ89 · 2023-08-18
> > **Thanks for the response.**
> >
> > Thanks to the authors for the response and additional clarifications of my questions. I understand that the significance of choosing two basic methods is to explore the distribution of the seed space. In addition, I accept the author's explanation that SD cannot generate visual images usually, and I hope that a brief explanation can be given in the final version. I have updated my rating accordingly.

---

> > > ### Author Response · Authors · 2023-08-20
> > > **Thank you for your review**
> > >
> > > Dear reviewer,
> > > Thank you for your support, for the productive discussion and for the insightful feedback that helped us improve the paper!
> > > We will provide the explanation in the final version as suggested.

---

### Official Review · Reviewer_WuD2 · 2023-07-05

**Soundness:** 2 fair
**Presentation:** 3 good
**Contribution:** 2 fair
**Rating:** 5
**Confidence:** 3

**Summary:**

The paper observed that the seed (noise) for the trained diffusion model has a property that the norm of the seed, which follows the $\chi$ distribution, is concentrated around a certain positive number, $\sqrt{d}$ where $d$ denotes the seed dimension. Based on this observation, the paper proposed a way to explore the seed space (e.g. interpolation, centroid) using Norm-Aware Optimization (NAO), which defines the objective function as likelihood maximization in the seed space. The paper applied seed space exploration with NAO to generate rare concepts and augment semantic data for few-shot classification and long-tail learning.

**Strengths:**

- The paper clearly stated contributions with comparisons with existing works on latent space exploration.
- The proposed method relies on the inherent structure of latent space, which is defined as the normal distribution, so could be used generally under diffusion model literature.
- Experiments are well-designed and easy to follow.

**Weaknesses:**

- Some experimental conditions (e.g. number of piece-wise linear paths) are unclear.
- Although the proposed method uses multiple approximations, the paper does not provide analysis or experiments on the accuracy of the approximation.
- Comparison for the rare-concept generation using centroid estimations seems not fair. Compared baseline is initialized randomly, but the proposed method is initialized with the Euclidean centroid.

**Questions:**

- It is unclear how many piece-wise linear paths are used to estimate (1) in experiments. Also, there is no discussion or analysis on $\delta$. If one set small $\delta$, the objective function (2) would be harder since the number of variables increases. If one sets large $\delta$, the approximation will be inaccurate. The paper mentioned that they constrain consecutive path points to be close (lines 161-162), but the corresponding description is unclear. Especially, in line 212, what is the meaning of the constraints, $c(x)\leq0$? There is no definition of the function $c(\cdot)$ and the input argument $x$.
- For the centroid estimation, there are at least two approximation gaps: one from discretizing the line-integral of log-likelihood (2) and the other from sub-optimal solution for (2). However, there is no analysis of the accuracy of the proposed distance function and centroid estimation, except for empirical performance whose accuracy seems to be dominated by the Euclidean centroid initialization.
- For the rare-concept generation, the paper claims that SeedSelect with the initial point found by NAO-centroid achieves faster and better generation. However, NAO-centroid also initializes its centroid with the Euclidean centroid of inversion points whereas SeedSelect uses random initialization. For a fair comparison, the runtime and performance of the SeedSelect initialized with the Euclidean centroid should be compared. Note that the Euclidean centroid initialization of the NAO-centroid is “to speed up convergence” (in line 201).
- In Table 4, CIFAR-FS $T_{Opt}$  might be 21 sec, not 21 min. And there is no unit for miniImageNet $\bar{T}_{Opt}$.

**Limitations:**

Yes, the authors describe the limitations of the proposed method.

---

> ### Author Rebuttal · Authors · 2023-08-09
>
> We sincerely appreciate your positive feedback, acknowledging the broad usability of our approach and the well-designed, easy-to-follow experiments.  We address your comments below.
>
> #### **Q1: It is unclear how many piece-wise linear paths are used.**
> **A1:**  For few-shot learning benchmarks we optimized paths with a length of 200 points between the centroid of the k-shot training samples and the samples themselves (see line 108 suppl. material). For long-tail learning, we optimized paths ranging in length from 10 to 200, adjusting based on the available number of samples for each class (line 121 suppl. material). Due to lack of space we present more details in Appendix F of the supplementary material.
>
> #### **Q2: What is the constraint c(x)≤0?**
> **A2:** $c(x)$ corresponds to the constraint in Eq. 2, namely $|x_i-x_{i-1}|  \leq \delta$. We formulate this constraint as $c_{i}(x) = |x_i-x_{i-1}| -\delta \leq 0$. This constraint is part of the integral discretization (into sum of finite elements) in Eq. 2. In the implementation we enforce this constraint using a soft penalty term with ReLU function; that is ReLU($c(x)$) (as described in line 212). Now there is a penalty for positive $c(x)$ (when the constraint is not satisfied). We apologize for the brevity in lines 211-212. We will elaborate on the discretization description of this equation in suppl. material and make sure $c(x)$ is clearly defined in text in the revised version.
>
> #### **Q3: Missing analysis for delta.**
> **A3**: We appreciate the reviewer's valuable feedback and their contribution to improving our paper. In our experiments, we set $\delta$ to be $||z_1 - z_2||/n$, where $z_1$ and $z_2$ are seed inversions of real images ($x_1$ and $x_2$), and $n$ is the number of interpolation points to be optimized. In response to the reviewer's comment, we conducted an in-depth analysis to investigate the impact of different delta values on the FID of images generated along the interpolation path. Figure G2 (see pdf attached to the general response to all reviewers) presents the results of this analysis. The analysis is done with $n=100$ points. The findings reveal that excessively low or high delta values adversely affect the FID, as they either constrain the points too closely together, causing overlap, or spread them too far apart, leading the path into low-likelihood regions.
>
> #### **Q4: Analysis of the accuracy of the proposed distance function.**
> **A4:** Indeed solving the optimization problem in Equation (2) provides only an approximation of its continuous counterpart in Equation (1).  We agree that there are two main error components: (i) approximation error of the discrete integral compared to the continuous one; (ii) the optimization error when optimizing Equation 2. This is an important observation, and we will include a discussion of it in the paper.
> With regards to (i), since $\log P(x)$ is smooth, we expect the minimizer of Equation (1) to be smooth as well and this allows piecewise linear paths to approximate it with arbitrarily small error by using enough point in Equation (2).
> Regarding (ii), we refer to this concern in lines 213-214. As is typically done in deep learning, we employ first-order optimization methods to optimize a non-smooth function. Therefore, analyzing the error and convergence is a very difficult problem that is beyond the scope of this paper. Based on our experimental results and 2D visualizations, we believe that our optimization process converges to a satisfactory solution. From a practical standpoint, we have observed that although the numerical solutions may not be exact, the FID measures, which indicate the quality of the generated images, consistently demonstrate high quality. This is also evident from the example images displayed in Figure 3 of the main paper and Figure S3 in the supplementary material.
>
> #### **Q5: Comparison with SeedSelect initialized with the Euclidean centroid.**
> **A5:** We indeed compare our results with SeedSelect initialized with Euclidean centroid and also with the Normalized Euclidean centroid, for fair comparison. This is shown in Table 1 (main paper, Page 7, L3,5). The results indicate that the performance improvement achieved by NAO is not solely attributed to the Euclidean initialization, but rather to the optimization minimization problem formulated in Equation (2) of our paper. Actually, initiating centroids with methods other than NAO, hurts SeedSelect performance (see Table 1). In terms of classifier accuracy, *NAO+SeedSelect* outperforms *Euclidean+SeedSelect* and *Normalized Euclidean + SeedSelect* by +94.3% and +12.3%, respectively. Additionally, in terms of FID score, *NAO+SeedSelect* performs +12.8% better than *Normalized Euclidean + SeedSelect*. For few-shot and long-tail benchmarks, we employed the best strategy as identified in Table 1, namely, *NAO+SeedSelect*.
>
> #### **Q6: Typo in Table 4.**
> **A6:** Yes, it should have been 21 sec. Thank you. Will be fixed in the camera-ready version.

---

> > ### Comment · Reviewer_WuD2 · 2023-08-17
> > **Thanks for the response.**
> >
> > I appreciate the response from the authors. I apologize for the instances that I missed during the reviewing of the paper and most of my concerns about the lack of experimental conditions are adequately resolved. Also, I'm pleased to hear that the authors will consider including a discussion about the approximation accuracy.
> >
> > My final question is about the Q5. The reason I thought that the SeedSelect is randomly initialized for *Rare-concept generation* is the lines 255-256.
> >
> > > SeedSelect [43] is a baseline method where a seed is randomly sampled and *no centroid is calculated*.
> >
> > Can authors clarify this? Does *no centroid is calculated* mean Euclidean centroid?

---

> > > ### Author Response · Authors · 2023-08-17
> > > **Euclidean Centroid with Seed Select**
> > >
> > > We appreciate your note. Table 2 shows the results from SeedSelect initialized with different methods, *random* in the first row and *Euclidean centroid* (Euclidean+SeedSelect), in the third row. The statement in line 255-256 is misleading. We apologize for that. We will correct this in the revised version.
> > > We’ll be glad to respond to any other concerns that you may have.

---

> > > > ### Comment · Reviewer_WuD2 · 2023-08-18
> > > > **Thanks for the timely response.**
> > > >
> > > > Thanks for the clarification from the authors. I understand that they conducted fair comparisons and found that the remaining concerns could be readily resolved by revising the manuscript. Accordingly, I updated my score to borderline accept.

---

> > > > > ### Author Response · Authors · 2023-08-20
> > > > > **Thank you for your review**
> > > > >
> > > > > Dear reviewer,
> > > > > Thank you for your support, for the productive discussion and for the insightful feedback that helped us improve the paper!
> > > > > We will make the clarification in the final version as suggested.

---

### Official Review · Reviewer_6Fzu · 2023-07-05

**Soundness:** 3 good
**Presentation:** 3 good
**Contribution:** 3 good
**Rating:** 6
**Confidence:** 4

**Summary:**

This paper investigates a new method for interpolating in the seed space of diffusion models, which is the Gaussian distribution used to initialize the generation process. Experiments demonstrate that diffusion models struggle with generation when the norm of the input differs from the distribution of norms of random noise samples drawn from the starting distribution; this can happen if the input is a LERP or SLERP interpolation between two random samples. The paper proposes to define a prior over the seed space using a chi distribution, and finds a path such that the likelihood of each point on the path is maximized under the prior distribution. This is optimized in a discretized fashion. A similar method can be used to find the centroid between multiple images, which minimizes the likelihood of the paths from each latent to the centroid. These methods are used to generate additional data, and studied in the context of rare-concept generation, few-shot recognition, and long-tail recognition.

**Strengths:**

- The proposed method is demonstrated to generate effective images in limited data scenarios. By providing a few example images, the model can generate related images by interpolating between the inputs under high likelihood regions of the latent space.
- The method is effective when combined with prior existing methods. SeedSelect optimizes an initial seed to match the concepts in a few given images. When using the NAO-centroid to initialize the seed, the results are more effective and optimization is faster compared to using SeedSelect alone.
- Qualitatively, the results look compelling against other shown interpolations in the diffusion seed space.

**Weaknesses:**

- It seems that the investigation is only performed on the input latent space, but I'm curious if feature interpolation in alternative latent representations would preserver a stronger image prior. For example, Asyrp[1] demonstrates that smooth changes can be obtained by manipulating the h-space of a diffusion model. I think this would be a worthwhile baseline to compare to, rather than just interpolations in the input latent space.
- The method relies on optimization over over a set of points. I think more details on the optimization could be provided here -- for example, how do the results differ if the number of interpolation points is changed, or the optimization time changes? What is the variation in this optimization procedure?

[1] Asyrp: https://arxiv.org/abs/2210.10960

**Questions:**

- in L212, what are the constraints $c(x) \geq 0$?
- Figure 1 left was difficult to understand in the first pass. Perhaps it would help to make clearer that the color refers to log likelihood of the $\chi$ distribution?
- Table 4: should it be 21 seconds rather than minutes?
- Are the optimization times stated per image in Table 4?
- What is the total overall overhead for the long-tail experiments -- how many images are generated per class?

**Limitations:**

Limitations are addressed in the conclusion. The key limitation is that this method requires additional optimization to generate one image, on the order of 30-60 seconds. The most effective use case of this method seems to be in conjunction with SeedSelect to produce plausible images with less optimization time, as the NAO method alone does not always produce recognizable images, as shown in Table 2.

---

> ### Author Rebuttal · Authors · 2023-08-09
>
> Thank you for finding our approach effective with compelling results. We address your comments below.
>
> #### **Q1: Comparing with Asyrp [1].**
> **A1**: We value the reviewer's suggestion to conduct a comparison between interpolation in the input space (our approach) and interpolation in a feature space (Asyrp). After a deep look, we found that a direct comparison is not feasible due to several reasons. First, The h-space lack properties of interpolation as it is primarily designed for editing of a predefined set of concepts. Second, the h-space edits carried out at each denoising step complicate the prospect of straightforward image interpolation between images in a single space. Despite these challenges, we attempted to interpolate (using different interpolation techniques and NAO) between h tensors of two images through all denoising steps. The outcome yielded corrupted images, underscoring the unsuitability of h-space for effective interpolations.
>
> #### **Q2: More details regarding optimization.**
> **A2:** Thank you for your valuable feedback. In response to this comment, we conducted additional experiments, replicating those performed in Table 1 of the main paper. We evaluated our approach by gradually increasing the number of optimization points and analyzed its impact on both FID scores and convergence time. The results demonstrate that an increase in the number of points leads to slightly improved FID scores, with only a marginal rise in optimization time. We will add this experiment and additional details regarding the optimization procedure in our revised manuscript/suppl. material. Variational analysis will be added too.
>
>
> |                             |  NAO-path |                |   NAO-centroid |           |
> | -----------            |  ----------- | -----------  |  ----------- | ----------- |
> |    #points            | FID (lower is better)           | $T_{init}$ (lower is better)| FID (lower is better)           | $T_{init}$ (lower is better) |
> |     10                   | 6.78+-0.1 | 21+-1 s | 5.48+-0.15 | 26+-1s |
> |       50                 | 6.7+-0.09| 23+-1 s|  5.47+-0.11 | 26+-1s |
> |         100             | 6.67+-0.06 | 24+-1 s | 5.43+-0.10 | 28+-1s|
> |         1000           | 6.61+-0.05| 27+-1 s| 5.41+-0.08| 30+-1s        |
>
> #### **Q3: What is the constraint c(x)≤0?.**
> **A3:** $c(x)$ corresponds to the constraint in Eq. 2, namely $|x_i-x_{i-1}|  \leq \delta$. We formulate this constraint as $c_{i}(x) = |x_i-x_{i-1}| -\delta \leq 0$. This constraint is part of the integral discretization (into sum of finite elements) in Eq. 2. In the implementation we enforce this constraint using a soft penalty term with ReLU function; that is ReLU($c(x)$) (as described in line 212). Now there is a penalty for positive $c(x)$ (when the constraint is not satisfied). We apologize for the brevity in lines 211-212. We will elaborate on the discretization description of this equation in suppl. material and make sure $c(x)$ is clearly defined in text in the revised version.
>
> #### **Q4: Figure 1 was difficult to understand.**
> **A4:** We thank the reviewer for their feedback. As pointed out by the reviewer, the colors in the figure correspond to the log-likelihood of the chi distribution. To address this concern, we will provide additional clarification on the figure and include more detailed explanations in the text for the camera-ready version.
>
> #### **Q5: Typo in Table 4.**
> **A5:** Yes, a typo,  it should be 21 sec. Thank you. Will be fixed in the camera-ready version.
>
> #### **Q6: Are the optimization times stated per image in Table 4?**
> **A6:** Yes, for a fair comparison, we provided the optimization time per image when employing SeedSelect for rare concepts. Notably, our approach significantly reduces the optimization time from 5 minutes to just ~25 seconds and also leads to reduced memory requirements, allowing concurrent generation of multiple images. It is important to mention that for common (head) concepts, optimization is *unnecessary*, and our approach enables direct image generation.
>
> #### **Q7: How many images are generated per class for long-tail learning?**
> **A7:**  We followed the experimental protocol of  [2] and generated samples for each class until the combined total of real and generated samples equaled the count of the class with the highest number of samples in the dataset, resulting in a uniform data distribution. For more details, please refer to Appendix F in the Supplementary material.
>
> #### **Q8: The method requires additional optimization to generate one image.**
> **A8:** This is correct for only rare concepts. For rare concepts, It requires additional optimization but NAO computational cost is significantly cheaper than the alternative, SeedSelect. While SeedSelect needs to backpropagate through the diffusion model NAO makes a simple path optimization process in the seed space. Please also refer to the response provided in the shared answer for all reviewers.
>
> #### **References**
> [1].   Kwon et al. (2023), "Asyrp: Diffusion Models already have a Semantic Latent Space".
> [2].   Samuel et al. (2023), "It's all about where you start: text to image generation with seed selection".

---

> > ### Comment · Reviewer_6Fzu · 2023-08-11
> > **Response to author rebuttal**
> >
> > Thanks to the authors for the response and additional clarifications. I agree with the other reviewers that the baselines presented are relatively simple, but I think the method is promising with regards to tasks in which data is limited by allowing a generator to produce additional samples from the limited data points (few-shot and long-tail applications). However, I find that improving the clarity of the paper would extremely helpful. I have updated my rating accordingly.

---

> > > ### Author Response · Authors · 2023-08-20
> > > **Thank you for your review**
> > >
> > > Dear reviewer,
> > > Thank you for your support, for the productive discussion and for the insightful feedback that helped us improve the paper!
> > > We will work hard to improve the clarity of the final version as suggested.

---

### Official Review · Reviewer_qXwY · 2023-07-11

**Soundness:** 3 good
**Presentation:** 3 good
**Contribution:** 3 good
**Rating:** 5
**Confidence:** 4

**Summary:**

This paper makes the observation that current training procedures make diffusion models biased toward inputs with a narrow range of norm values. To address this issue, the authors propose a novel method for interpolating between two seeds and demonstrate that it defines a new non-Euclidean metric that takes into account a norm-based prior on seeds. The authors describe a simple yet efficient algorithm for approximating this metric and use it to further define centroids in the latent seed space. The effectiveness of the proposed approach is validated on generating images of rare concepts, and augmenting semantic data for few-shot classification and long-tail learning.

**Strengths:**

1. The observation that current training procedures make diffusion models biased toward inputs with a narrow range of norm values is interesting and inspiring.
2. The proposed approach is well-motivated and aligns with intuition.
3. The observation and proposed approach have the potential to benefit many tasks related to the application of diffusion models.

**Weaknesses:**

1. The proposed approach only works well with seed optimization techniques such as SeedSelect, as indicated by Fig.5 and lines 195 - 203. This implies that the derived interpolation paths may not be optimal and it also introduces extra computational cost of the seed optimization.
2. The proposed approach is demonstrated mainly on generating rare images and augmenting data for few-shot learning. However, I am more interested in more applications that might benefit from the proposed approach. For example, can the proposed approach be applied for video interpolation and video generation and perform better than previous approaches?
3. Although significantly better than previous approaches, the results in Fig.3 indicate that the interpolation results of the proposed approach is still not perfect. Do the authors have insights on the reason for the imperfect interpolation results?

**Questions:**

Please refer to the weakness section.

**Limitations:**

The limitation is adequately addressed.

---

> ### Author Rebuttal · Authors · 2023-08-09
>
> Thank you for finding our approach inspiring and interesting. We address your comments below.
>
> #### **Q1: The proposed approach only works well with seed optimization.**
> **A1:** Let us explain where and why NAO works well without seed optimization and the relation between the two methods. First, we stress that seed optimization is only necessary for generating rare concepts (See SeedSelect [1] and in lines 29 and 62 of our paper). For rare concepts, diffusion models often fail. They may generate high-quality images but from a wrong category. Seed optimization presents a solution for this scenario.
>
> NAO functions effectively without relying on seed optimization. This is demonstrated in Figures 1, 3, S3, and Table 1. NAO's interpolations and centroids enable the generation of semantically consistent images even for **common (head)** concepts where SeedSelect is unnecessary. For instance, head classes like "tuna" and "jersey" have multiple interpretations and can lead to the generation of incorrect objects. "Tuna" might denote either a "tuna fish" or a "tuna can," while "jersey" could refer to a "standard t-shirt" or a "sports team shirt".  Given a few reference images with their concept name, NAO can generate images that belong to the correct domain.
>
> To further illustrate NAO’s advantage, we replicated the experiments from Table 1 solely on 50 randomly selected **common (head)** concepts, **without employing seed optimization**. Subsequently, we compared NAO to two alternate interpolation methods in the provided table. The results unequivocally confirm that NAO achieves significantly higher accuracy (in generating the correct concept) and improved FID.
>
> |                             | Acc (higher is better)           | FID (lower is better)|
> | -----------            |  ----------- | ----------- |
> | LERP                  | 0.00      | 60.60       |
> | SLERP                | 69.12   | 17.55        |
> | **NAO-path (ours)**| **88.91**   | **5.21**        |
>
> #### **Q2: Can the proposed approach be applied for video interpolation and video generation?**
> **A2:** We found the idea of applying our method to video interpolation intriguing, following the reviewer's suggestion. As a result, we applied our approach to the task of video frame interpolation, aiming to generate an intermediate frame between existing consecutive frames (I1 and I2) in a video sequence. Using NAO, we interpolated between the seed inversions of two given frame images, optimizing 50 points, and generating an image from the middle points. Example images in Figure G1 (see pdf attached to the general response to all reviewers) illustrate NAO's ability to create intermediate frames. To further demonstrate NAO’s capability, we conducted also a quantitative evaluation. To this end, we followed the experiment outlined in [2,3], focusing on the **DAVIS dataset**: a widely acknowledged benchmark for Video Frame interpolation tasks. The evaluation of predicted frames against the ground truth was carried out using established metrics like PSNR and SSIM. The table demonstrates that our interpolation approach achieves comparable results to existing methods specifically tailored for video interpolation, despite solely using a pre-trained text-to-image model. We appreciate the reviewer's insightful suggestion, as this presents a promising avenue for future research.
>
> |                             | PSNR   (higher is better)        | SSIM  (higher is better)|
> | -----------            |  ----------- | ----------- |
> | MCVD [2]                  | 18.646      | 0.705       |
> | LDMVFI  [3]              | 25.541   | 0.833       |
> | **NAO-path (ours)**| 25.413  | 0.813        |
>
>
> #### **Q3: Do the authors have insights on the reason for the imperfect interpolation results?**
> **A3:** As indicated in A1 and also mentioned in [1], rare concepts often have restricted regions within the seed space where plausible and correct images can be generated. This is evident in Figure 3, where NAO performs well for the common concept of a jeep, but for the rare concept of a tiger cat, *most* images appear satisfactory. Despite this complexity, our approach surpasses existing interpolation methods by consistently generating superior, semantically accurate images at all interpolation points.
>
> #### **References**
> [1].   Samuel et al. (2023), "It's all about where you start: text to image generation with seed selection".
> [2].   Danier et al. (2023), "LDMVFI: Video Frame Interpolation with Latent Diffusion Models".
> [3].   Voleti et al. (2022), "MCVD: Masked Conditional Video Diffusion for Prediction, Generation, and Interpolation".
> [4].   Perazzi et al. (2016), "A benchmark dataset and evaluation methodology for video object segmentation".

---

> > ### Comment · Reviewer_qXwY · 2023-08-18
> > **Thank authors for the rebuttal**
> >
> > Thank the authors for the rebuttal. The authors have addressed most of my concerns so I updated my rating accordingly.

---

> > > ### Author Response · Authors · 2023-08-20
> > > **Thank you for your review**
> > >
> > > Dear reviewer,
> > > Thank you for your support, for the productive discussion and for the insightful feedback that helped us improve the paper!

---

### Author Rebuttal · Authors · 2023-08-08

Dear Reviewers and ACs,
We were happy to see that the reviewers have found our approach **"interesting and inspiring"**, **"well-motivated" (R1)**, and recognized its **potential to benefit various tasks related to diffusion models' applications (R1, R3)**. Additionally, they have acknowledged that our method is based on **"good theoretical support" (R4)** and have found our experiments to be **“well-designed”**, **“easy-to-follow” (R3)** and **"compelling" (R2)**.
We have addressed the reviewer's concerns in our rebuttal and are open to further discussion. Your input has been instrumental in improving our paper.


### General response to all reviewers

NAO presents an innovative method to interpolate between seeds, revealing a new non-Euclidean metric that takes into account a prior over samples. Our efficient algorithm approximates this metric, facilitating centroid estimation in the latent seed space and enhancing the generation of rare concept images.

#### **Q1: The proposed approach only works well with seed optimization.**
**A1:** Let us explain where and why NAO works well without seed optimization and the relation between the two methods. First, we stress that seed optimization is only necessary for generating rare concepts (See SeedSelect [1] and in lines 29 and 62 of our paper). For rare concepts, diffusion models often fail. They may generate high-quality images but from a wrong category. Seed optimization presents a solution for this scenario.

NAO functions effectively without relying on seed optimization. This is demonstrated in Figures 1, 3, S3, and Table 1. NAO's interpolations and centroids enable the generation of semantically consistent images even for **common (head)** concepts where SeedSelect is unnecessary. For instance, head classes like "tuna" and "jersey" have multiple interpretations and can lead to the generation of incorrect objects. "Tuna" might denote either a "tuna fish" or a "tuna can," while "jersey" could refer to a "standard t-shirt" or a "sports team shirt".  Given a few reference images with their concept name, NAO can generate images that belong to the correct domain.

To further illustrate NAO’s advantage, we replicated the experiments from Table 1 solely on 50 randomly selected **common (head)** concepts, **without employing seed optimization**. Subsequently, we compared NAO to two alternate interpolation methods in the provided table. The results unequivocally confirm that NAO achieves significantly higher accuracy (in generating the correct concept) and improved FID.

|                             | Acc (higher is better)           | FID (lower is better)|
| -----------            |  ----------- | ----------- |
| LERP                  | 0.00      | 60.60       |
| SLERP                | 69.12   | 17.55        |
| **NAO-path (ours)**| **88.91**   | **5.21**        |


#### **Q2: What is the constraint c(x)≤0?.**
**A2:** $c(x)$ corresponds to the constraint in Eq. 2, namely $|x_i-x_{i-1}|  \leq \delta$. We formulate this constraint as $c_{i}(x) = |x_i-x_{i-1}| -\delta \leq 0$. This constraint is part of the integral discretization (into sum of finite elements) in Eq. 2. In the implementation we enforce this constraint using a soft penalty term with ReLU function; that is ReLU($c(x)$) (as described in line 212). Now there is a penalty for positive $c(x)$ (when the constraint is not satisfied). We apologize for the brevity in lines 211-212. We will elaborate on the discretization description of this equation in suppl. material and make sure $c(x)$ is clearly defined in text in the revised version.

#### **References**
[1]    Samuel et al. (2023), "It's all about where you start: text to image generation with seed selection".

---

### Decision · Program_Chairs · 2023-09-21

**Decision:**

Accept (poster)

**Comment:**

All four reviewers were in favor of paper acceptance after author rebuttals. The paper provides an interesting new approach exploring/understanding of the latent space of text-to-image models.